# Excited State Kinetics of Benzo[a]pyrene Is Affected by Oxygen and DNA

**DOI:** 10.3390/molecules28135269

**Published:** 2023-07-07

**Authors:** Yunxia Han, Xueli Wang, Xiaoxiao He, Menghui Jia, Haifeng Pan, Jinquan Chen

**Affiliations:** 1State Key Laboratory of Precision Spectroscopy, East China Normal University, Shanghai 200241, China; 2Collaborative Innovation Center of Extreme Optics, Shanxi University, Taiyuan 030006, China

**Keywords:** benzo[a]pyrene, oxygen, excited state dynamics, transient absorption spectroscopy, time-resolved fluorescence

## Abstract

Benzo[a]pyrene is a widespread environmental pollutant and a strong carcinogen. It is important to understand its bio-toxicity and degradation mechanism. Herein, we studied the excited state dynamics of benzo[a]pyrene by using time-resolved fluorescence and transient absorption spectroscopic techniques. For the first time, it is identified that benzo[a]pyrene in its singlet excited state could react with oxygen, resulting in fluorescence quenching. Additionally, effective intersystem crossing can occur from its singlet state to the triplet state. Furthermore, the interaction between the excited benzo[a]pyrene and ct-DNA can be observed directly and charge transfer between benzo[a]pyrene and ct-DNA may be the reason. These results lay a foundation for further understanding of the carcinogenic mechanism of benzo[a]pyrene and provide insight into the photo-degradation mechanism of this molecule.

## 1. Introduction

As a typical polycyclic aromatic organic pollutant and a strong carcinogen, benzo[a]pyrene (BaP) is photo-toxicity and photo-genotoxicity, which is a serious health hazard to organisms after absorbing light energy in the ultraviolet region (320–400 nm) [1,2,3]. At the same time, photo-degradation is one of the main decomposition ways for this molecule. A study of its excited state dynamics is of great significance to further understand the bio-toxicity and degradation mechanism of benzo[a]pyrene.

In recent years, studies on the photo-degradation mechanisms of benzo[a]pyrene involving its excited state have been reported. Based on the generation of singlet oxygen and superoxide anion in benzo[a]pyrene aqueous solution, a photo-chemical degradation model related to sensitization of triplet excited states was proposed [4,5]. In addition, Yang et al. proposed the self-sensitive photo-chlorination mechanism of benzo[a]pyrene in acetonitrile and NaCl mixed aqueous solution by detecting photoreaction products under different irradiation times. They believe that the triplet states of benzo[a]pyrene formed by excited singlet states through intersystem crossing lead to the formation of [^3^BaP*-^3^O_2_] or [BaP-^1^O_2_] oxygen-containing complex, which can further react with Cl^−^ to form 6-ClBaP [6]. Meanwhile, benzo[a]pyrene can be inserted between two adjacent DNA base pairs as a typical DNA intercalation molecule [7], leading to local changes in DNA structure such as unwrapping and extension of DNA double strands. These structural modifications may lead to delay or inhibition of DNA transcription and replication, affecting normal physiological functions [8,9]. At the same time, UVA radiation and benzo[a]pyrene can synergically induce oxidative damage of DNA and significantly increase the single strand break rate of DNA [10,11,12,13,14]. A related photodamage mechanism suggests that benzo[a]pyrene in its high energy excited state may react with DNA to produce 8-hydroxy-2′-deoxyguanosine (8-oxodG) after intercalation into DNA.

It should be noted that most of the reaction mechanisms related to the excited state of BaP are derived from the inference of intermediate chemical products, but direct evidence of excited state dynamics is still lacking. Relevant quantum chemical calculations show that benzo[a]pyrene can go through intersystem crossing to triplet states after photoexcitation, and then undergo direct energy or electron transfer with oxygen to produce singlet oxygen or superoxide radical [15]. Blough et al. observed its triplet excited state absorption signals using a laser flash photolysis technique under 355 nm excitation, providing intuitive evidence for the existence of triplet states in benzo[a]pyrene [16]. Moreover, Banerje et al. studied the photophysical properties of benzo[a]pyrene molecules as well as the energy resonance transfer between benzo[a]pyrene and a variety of acceptor materials by steady-state and picosecond resolved fluorescence spectroscopy [17]. Yet, a complete excited state relaxation model for benzo[a]pyrene has not been established. Herein, we studied the excited state dynamics of benzo[a]pyrene and illustrate how oxygen and DNA could affect its excited state relaxation process by using time-resolved fluorescence spectroscopy and femtosecond to nanosecond time-resolved transient absorption spectroscopy. 

## 2. Results

### 2.1. Steady-State Absorption and Emission Spectra

Steady-state absorption and emission spectra of benzo[a]pyrene in acetonitrile are presented in Figure 1a. Steady-state absorption and fluorescence spectra show mirror symmetry [17]. The three well-characterized emission peaks at 403 nm, 427 nm and 454 nm along with a weak band near 485 nm match well with previous reports and they are arisen from different vibrational states in the S_1_ state of benzo[a]pyrene [7,18]. The radiative decay rate constant is 1.58 × 10^7^ s^−1^ based on a Strickler–Berg analysis (detail in Appendix A). Temperature-dependent emission spectra of benzo[a]pyrene were also recorded under air-saturated conditions (Figure 1b) and it shows a steady increase of the emission intensity as temperature decreases.

### 2.2. Femtosecond to Nanosecond TA Spectra of Benzo[a]pyrene 

To reveal the excited state dynamics of benzo[a]pyrene, femtosecond to nanosecond TA spectra were measured. Figure 2 depicts femtosecond TA spectra of benzo[a]pyrene following excitation at 380 nm. Immediately after excitation, two negative bands are observed at 345 nm and 363 nm and they are in excellent agreement with peaks in the steady-state absorption spectra. Meanwhile, a broad excited state absorption (ESA) band with three main peaks at 422 nm, 531 nm and 650 nm appears within the instrumental response time (~120 fs). Then, all ESA and ground-state bleach (GSB) signals decay together and are accompanied by the emergence of a new ESA band at 462 nm in the first 7 ns time window.

Nanosecond TA spectra of benzo[a]pyrene in acetonitrile were recorded and shown in Figure 3. Upon 380 nm excitation, a broad ESA signal ranges from 400 nm to 700 nm with three peaks at 422 nm, 531 nm, 650 nm as well as a shoulder at 462 nm, appears within the instrumental response time. Then, the ESA band at 462 nm keeps increasing until ∼24 ns while other peaks decay. After that, the whole TA spectra decay to baseline with no further spectra evolution. Global analysis of the TA spectra yielded two-time constants (11.4 ± 0.1 ns and 134 ± 1 ns). Decay-associated difference spectra (DADS) were extracted from a global analysis of the broadband TA data. The first DADS (cyan) exhibit a negative peak that is centered at 462 nm and 3 positive peaks located at 420 nm, 530 nm and 650 nm. It evolves into the second DADS with a lifetime of 11 ns. The second DADS (yellow) displays a broad positive band with a maximum at 462 nm and it has a lifetime of 134 ns. A sensitization experiment was carried out to verify whether these two components arise from triplet states (Appendix A). Platinum octaethylporphyrin (PtOEP) is a suitable sensitizer as its lowest triplet state has an energy of 1.92 eV (higher than the energy of benzo[a]pyrene’s T_1_ state) [19] and it can be selectively excited. As presented in Appendix A, the kinetics of PtOEP is significantly quenched in the presence of benzo[a]pyrene and the decay of the ESA signal centered at 415 nm corresponds to the rise of a new ESA band at ∼460 nm. The final spectral shape for the PtOEP and benzo[a]pyrene mixture is almost identical to the spectral shape of DADS2 in benzo[a]pyrene. Representative kinetics at 422 nm and 462 nm under air- and N_2_-saturated conditions are shown in Figure 3d,e. It is clear that the kinetics at 427 nm decay faster in air-saturated conditions compared with that in N_2_-saturated conditions. Meanwhile, the kinetics at 462 nm shows a build-up process in the first 200 ns under nitrogen conditions while this process is absent under air-saturated conditions. 

### 2.3. Femtosecond Transient Absorption Spectra of the Mixture of Benzo[a]pyrene and ct-DNA

Femtosecond TA measurements were also carried out for both ct-DNA and the mixture of benzo[a]pyrene and ct-DNA. As shown in Figure 4a, for ct-DNA, a broad ESA signal appears immediately in the whole probe range with two positive peaks centered at 345 nm and 550 nm upon 267 nm excitation. In the first 600 fs, the ESA signal at 550 nm increases slightly in amplitude while the TA signal around 350 nm decays. Then, the whole spectra decay to baseline within 330 ps. Global analysis of TA spectra for ct-DNA yields three lifetimes and the DADS are also shown in Figure 4a. The first DADS (black) has a positive peak centered at 355 nm as well as a negative band with a maximum of 530 nm. The second DADS (red), which shows a broad ESA band in the whole probe range has a lifetime of 4.3 ps. The lifetime of the third DADS (blue) is determined to be 45.8 ps and its spectral shape is similar to that of the second DADS. When benzo[a]pyrene was added to ct-DNA, its spectral evolution was almost the same as that of ct-DNA, and three lifetimes were determined to be 48.3 ± 6.0 fs, 3.7 ± 0.1 ps and 41.6 ± 0.8 ps, respectively (Figure 4b).

### 2.4. Time-Resolved Fluorescence Spectra

Figure 5a,b exhibits emission spectra and kinetics at 427 nm of benzo[a]pyrene under air- and N_2_-saturated conditions. It is clear that both the emission intensity and lifetime of benzo[a]pyrene are significantly quenched in the presence of oxygen. Furthermore, a comparison of the steady-state fluorescence emission spectra and the fluorescence kinetics of benzo[a]pyrene in the presence and absence of ct-DNA are displayed in Figure 5c,d, respectively. It is clear that the addition of ct-DNA results in a 7 nm red-shift of the fluorescence emission peak compared to benzo[a]pyrene monomer. In addition, the fluorescence decay of benzo[a]pyrene itself can be well-fitted by a mono-exponential function, yielding a lifetime of 11 ± 0.1 ns. However, after adding ct-DNA, it is now required a three-exponential function to fit the kinetics perfectly and the lifetimes are 0.38 ± 0.2 ns, 2.1 ± 0.1 ns and 8.2 ± 0.1 ns, respectively.

## 3. Discussion

### 3.1. The Fluorescence Lifetime of Benzo[a]pyrene Is Affected by Oxygen

Global fitting of the TA spectra of benzo[a]pyrene yielded two lifetimes and DADS are shown in Figure 3c. The 134 ns lifetime is clearly demonstrated to arise from the lowest triplet excited state (T1) of benzo[a]pyrene from the sensitization experiments. The triplet quantum yield of benzo[a]pyrene was measured to be 0.23 ± 0.02 (detail in Appendix A). On the other hand, the 11.4 ns component seen in the TA spectra matches well with a previous study on benzo[a]pyrene [17]. In addition, the DADS1 (green) in Figure 3c clearly exhibits a negative dip around 460 nm, which should be due to the stimulated emission and suggests that there is a transition between this component and the 134 ns component. The 11.4 ns component was also detected in the TCSPC measurement, indicating that this is a luminous state. However, in nitrogen-saturated conditions, the emission of benzo[a]pyrene is enhanced (Figure 4a) and the lifetime of this component increases to 25 ns (Figure 4b). The fluorescence quantum yield of benzo[a]pyrene in air is 0.15 ± 0.01 by absolute method while its fluorescence quantum yield increased to 0.38 ± 0.02 after deoxygenation. Indeed, oxygen-sensitive fluorescence emission has been reported in benzo[e]pyrene [20], an analogue of benzo[a]pyrene. Yet, no specific explanation has been given in previous studies. We envisage that either thermally activated delayed fluorescence (TADF) or direct fluorescence quenching by oxygen could lead to this phenomenon and a detailed discussion is below. 

In order to verify if there is TADF in benzo[a]pyrene, temperature-dependent emission spectra were measured from 137 K to 237 K (Figure 1b). It is clear that the luminescence intensity decreases continuously with the increase in temperature while the emission wavelength over temperature remains unchanged. Therefore, TADF is excluded in benzo[a]pyrene. 

Meanwhile, it can be estimated that the collision time between oxygen and benzo[a]pyrene through diffusion is approximately 13.6 ns based on the solubility of oxygen in acetonitrile and the diffusion rate constant [21]. As the lifetime of the emissive singlet state of benzo[a]pyrene is 25 ns in nitrogen-saturated conditions, it is highly possible that the singlet excited state of benzo[a]pyrene can be quenched by oxygen. Actually, similar phenomena have also been reported in molecules such as fluorescein, pyrrolidine B and pyrrodine Y [21,22,23]. Moreover, fluorescence lifetimes and intensity of molecules similar to benzo[a]pyrene (naphthalene, phenanthrene, chrysene and pyrene) in cyclohexane solution were reported to be dependent on the oxygen concentration [24]. In addition, as shown in Appendix A, the absorption spectra before and after deoxygenation barely change and they also match well with the fluorescence excitation spectra. These results also exclude the possibility of the ground state of BaP forming a complex with oxygen resulting in the static quenching of fluorescence. Thus, we believe that the reaction between benzo[a]pyrene and oxygen takes place in the excited state rather than in the ground state, leading to the observed fluorescence quenching effect.

Based on the above analysis, we propose the excited state relaxation mechanism of benzo[a]pyrene in acetonitrile (Figure 1). After excitation, benzo[a]pyrene should initially populate the ^1^ππ* state in the Franck–Condon (FC) region with excess vibrational energy. After conformational and solvent dynamics, the excited state population is trapped in the ^1^ππ* minimum and then either return to the ground state or intersystem cross to the triplet state. The population in the ^1^ππ* state could be quenched by the oxygen in the solution and it is a diffusion control process. 

### 3.2. The Fluorescence Lifetime of Benzo[a]pyrene Is Affected by ct-DNA

Some typical polycyclic aromatic hydrocarbon (PAHs) such as acenaphthylene, acenaphthene, and middle cyclic PAHs (fluoranthrene, pyrene, benzo[a]anthracene and acenaphthene) all exhibited fluorescence quenching after binding to DNA. The quenching rate constants of these PAHs binding to DNA are 3~5 orders of magnitude higher than the maximum *K*q (bimolecular quenching rate constant) value of diffusion-controlled quenching process. Therefore, in addition to dynamic quenching, static fluorescence quenching also exists during the fluorescence quenching process between PAHs and DNA [25]. Fluorescence quenching also occurs in two pyrene derivatives, 1-aminopyrene and 1-pyrenebutylamine, after their intercalation with DNA. Both steady-state and time-resolved fluorescence spectra indicate that they are static quenching [26]. In our study, steady-state fluorescence emission spectra and time-resolved fluorescence spectra were used to study the interaction between benzo[a]pyrene and DNA. Steady-state fluorescence emission spectra and fluorescence decay spectra of benzo[a]pyrene in the present and absent of ct-DNA indicate that ct-DNA can also quench the fluorescence of benzo[a]pyrene. Three time constants (0.38 ± 0.2 ns, 2.1 ± 0.1 ns and 8.2 ± 0.1 ns) were determined in benzo[a]pyrene and ct-DNA complex and this lifetime matches well with a previous report by Banerjee and co-workers [27], in which they conclude that fluorescence quenching is due to charge transfer between benzo[a]pyrene and ct-DNA. In this study, we tried to capture the charge transfer process between benzo[a]pyrene and ct-DNA by using femtosecond TA spectroscopy. Unfortunately, there is no significant difference in the TA spectra of ct-DNA in the presence or absence of benzo[a]pyrene upon 267 nm excitation as shown in Figure 4. This could due to either the very low solubility of benzo[a]pyrene or the TA signal arise from charge transfer is too small to be detected. 

Nevertheless, oxidative DNA damage is a well-known mechanism in DNA. 8-oxodG is the most common oxidation-generating lesion, which is closely related to mutation and carcinogenesis. Guanine is the most easily oxidized nucleic acid base, so it is attacked by most oxidants and can cause specific DNA double-strand breaks that cause DNA damage. It has been reported that benzo[a]pyrene can be electro-oxidized to benzo[a]pyrene-2OH compounds at 1.2 V vs. Ag/AgCl [28]. Meanwhile, the redox potential of dGMP is about 0.85 V, which suggests that benzo[a]pyrene could draw an electron from the guanine base in DNA [26]. In addition, the triplet state of benzo[a]pyrene also can react with oxygen to form singlet oxygen. Guanine is known to react with ^1^O_2_ to form endoperoxide, which further forms 8-oxoGua. Indeed, this oxidative damage of benzo[a]pyrene to DNA was confirmed by previous studies in which 8-oxoGua products were identified and double-strand break of DNA was demonstrated in both cell-free system (in vitro) and cultured Chinese hamster ovary (CHOK1) cells [11,29]. Yet, direct observation of electron transfer between benzo[a]pyrene and DNA has not been reported. From our experimental results, it suggests that we have to improve the signal-to-noise ratio of the TA setup due to the low solubility of benzo[a]pyrene even after it binds with ct-DNA.

## 4. Materials and Methods

### 4.1. Chemicals

Benzo[a]pyrene was purchased from Innochem (Shanghai, China). Spectral-grade acetonitrile was purchased from Aladdin (Shanghai, China). 2-Methyltetra hydrofuran (2-MeTHF) was purchased from J&K Chemical Ltd. (Shanghai, China) and Platinum octaethylporphyrin (PtOEP) was purchased from Frontier Scientific (Beijing, China). The DNA used in the experiment was a highly polymerized calf thymus-DNA (ct-DNA) purchased from Sigma-Aldrich (Shanghai, China).

The aqueous solution was prepared in Tris-HCl buffer (100 mM Tris, 100 mM HCl, pH = 7.4). The benzo[a]pyrene–DNA complexes were made at room temperature by mixing fine crystals of benzo[a]pyrene with ct-DNA in Tris-HCl buffer. After 48 h agitation, the mixture solution was centrifuged for 20 min in order to remove excess supernatant. All samples were freshly prepared for each measurement.

### 4.2. Steady-State Absorption and Emission Spectroscopy Measurements

A double-beam UV–vis spectrophotometer (TU1901, Beijing Spectrograph General Instrument Co., Ltd., Beijing, China) and commercial fluorescence spectrometer (FluoroMax-4, HORIBA) were used to record steady-state absorption and emission spectra at room temperature, respectively. Samples were held in a quartz cuvette with a 2 mm optical length. The temperature-dependent fluorescence emission spectra from 277 K to 137 K were measured in 2-MeTHF by using Edinburgh Instruments Fluorescence Spectrometer FLS1000 fluorimeter (Edinburgh Instruments, Livingston, MT, USA). The temperature of the sample was controlled by an Oxford OptistatDN Cryostat (Oxford Instruments nanoscience, Oxford, UK).

### 4.3. Time-Resolved Transient Absorption (TA) Spectroscopy Measurements 

Femtosecond TA spectra were obtained using a femtosecond TA spectrometer (Helios, Ultrafast systems, Sarasota, FL, USA) [30,31,32]. The fundamental beam with a central wavelength of 800 nm, 90 fs pulse, and a repetition rate of 1 kHz, was generated by a Ti:sapphire laser system (Astrella, Coherent). Then, a fraction of the fundamental beam was used to produce a specific wavelength pump beam via an optical parametric amplifier (OPerA Solo, Coherent Inc., Santa Clara, CA, USA). Another part of the fundamental beam passed through a delay line and then focused on CaF_2_ or sapphire crystal to generate a white light continuum (WLC) probe beam. Nanosecond TA data were measured using a TA spectrometer (Helios-EOS fire, Ultrafast System, Sarasota, FL, USA). The pump beam was derived from the same Ti:sapphire amplifier as the femtosecond experiment. The WLC probe beam was generated from a photonic crystal fiber broadband probe source. TA spectra were analyzed by Igor Pro program version 6.21, OriginPro 2017C and Glotaran program version 1.5.1 [33]. 

### 4.4. Time-Correlated Single Photon Counting (TCSPC) System

Fluorescence lifetimes were measured on a TCSPC system. A picosecond super-continuum fiber laser (SC400-pp-4, Fianium, Eugene, OR, USA) generates the excitation pulse with a repetition rate of 10 MHz. Fluorescence was recorded on a TCSPC module (PicoHarp 300, PicoQuant, Berlin, Germany) and a microchannel plate PMT (R3809U-50, Hamamatsu Photonics, Shizuoka, Japan). A monochromator (7ISW151, Sofn Instruments, Beijing, China) was used to select the emission wavelength. The instrument response function (IRF) of this system was determined to be ∼200 ps by measuring the scattering of silica solutions.

## 5. Conclusions

In this work, we elucidate the excited state dynamics of benzo[a]pyrene in solution by using time-resolved fluorescence spectroscopy and femtosecond to nanosecond transient absorption spectroscopy. For the first time, we demonstrate that oxygen sensitivity of its fluorescence is due to the interaction between benzo[a]pyrene in its singlet excited state and oxygen rather than the formation of ground state oxygen complex or triplet state quenching. Compared with oxygen, ct-DNA could also quench the fluorescence of benzo[a]pyrene and a possible electron transfer pathway is proposed. Our findings provide new insights into the understanding of the photodegradation mechanisms of PAHs and their interaction mechanism with DNA. 

## Data Availability

Data will be made available on request.

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
