# Peer review of "Excited State Kinetics of Benzo[a]pyrene Is Affected by Oxygen and DNA"

_molecules, 2023, doi:10.3390/molecules28135269_

Round 1
Reviewer 1 Report
Referee report on the MS 2473551
“Excited State Dynamics of Benzo[a]pyrene Is Affected by Oxygen and DNA”
by Yunxia Han, Xueli Wang, Xiaoxiao He, Menghui Jia, Haifeng Pan, and Jinquan Chen
submitted to Molecules
Benzo[a]pyren (BaP) is a polycyclic aromatic hydrocarbon formed, for instance, in combustion processes. It is a well-characterized and potent carcinogen. Of course, the interaction of such a compound with DNA as well as its degradation pathways are of great interest. As BaP absorbs near UV radiation, also photo-induced processes play a crucial role here. The authors address this issue by time-resolved UV/Vis absorption as well as fluorescence spectroscopy. While I find the topic well suited for “Molecules” and also the applied methods appropriate, I see a couple of serious flows as listed below. Thus, the manuscript should be reconsidered after a major revision.
Major issues
1. In the femtosecond experiment on BaP and calf thymus DNA, the authors use 267 nm excitation. It is well known that around this wavelength, the absorption of DNA bases peaks (see e.g. C.T. Middleton et al., Annual Review of Physical Chemistry 60 (2009) 217-239). I assume that this experiment was conducted with an excess of DNA bases. (The authors fail to quantify that.) Under these circumstances, the DNA bases act as an inner filter “shielding” BaP from the excitation. It is, thus, no surprise that the DNA-only and the BaP/DNA experiments look very similar (Figure 4). The experiment should be repeated with the excitation tuned to 380 nm (peak of BaP absorption). At this wavelength DNA essentially does not absorb. Concerning the interpretation of the DNA-only experiment, the authors ought to have a look at the Middleton review.
2. Related to that: For the concentration of BaP and DNA (in terms of base pairs) employed, the authors ought to estimate the fraction of intercalated BaP (for an approach see, e.g. X.Y. Zhou, G.W. Zhang, L.H. Wang, Biological Macromolecules 67 (2014) 228-237). For a fraction close to one, it is clear that quenching of BaP by DNA happens in the static limit. This would greatly simplify the discussion (section 4.2.)
3. The authors fail to report important photophysical parameters of BaP:
- Peak absorption coefficients
- Fluorescence quantum yields in the presence and absence of oxygen
- Radiative rate constant based on a Strickler-Berg analysis
- Triplet quantum yield estimated using the ground state bleach
Minor issues
a) The term “dynamics” in the title and throughout the paper should be replaced by “kinetics”, see P. Kambhampati, The Journal of Physical Chemistry Letters 14 (2023) 2996-2999.
b) On page 3, the authors attribute the fluorescence to the “S2 state”. I guess this should read “S1 state”. If otherwise, this non-Kasha behavior should be elaborated on.
c) The DADS1 in Figure 2d seems to have minima at the maxima of the fluorescence spectrum. They could, thus, be due to stimulated emission. This should be commented on.
(d) There are DADS in Figures 2 and 3 and they look identical to me. This is confusing.
(e) The part in the Discussion on the oxygen quenching of the singlet excitation should be shortened. Such a quenching is textbook knowledge.
(f) On page 4, the authors state: “After that, the whole TA spectra decay to baseline within 820 ns.” This statement has the implication that the lifetime is 820 ns which is not the case.
Text should be proofread by a native speaker.
Reviewer 2 Report
The paper by Y. Han et al. reports an experimental study of the excited-state dynamics of benzo[a]pyrene in acetonitrile in the absence/presence of molecular oxygen and ct-DNA. By analyzing time-resolved fluorescence and transient absorption spectra, the authors conclude that fluorescence is dynamically quenched by oxygen in the singlet excited state of benzo[a]pyrene. ct-DNA is also observed to quench the fluorescence of benzo[a]pyrene presumably due to charge transfer in their complexes, which are formed in the ground electronic state. This paper is interesting and can be published in Molecules. However, there are several issues that have to be resolved prior to its publication:
1. It would be helpful to include a more elaborated discussion on the mechanism of fluorescence quenching by molecular oxygen. Is there any charge transfer involved, similar to ct-DNA?
2. In section 3.1, it is mentioned that the S2 state is populated in benzo[a]pyrene upon photoabsorption. However, scheme 1 displays the first singlet and triplet pipi* states. This has to be consistent. Furthermore, if the S2 state is initially populated, the authors should comment on the possible decay mechanism to the S1 state.
3. Figure 2d and Figure 3c show the decay associated difference spectra obtained from global fitting. The spectral decomposition with two time components show negative and positive signals at 462 nm. What is the physical meaning of the negative signal at 462 nm associated with a lifetime of 11.4 ns?
4. What is the origin of the band at 462 nm?
5. Finally, the authors use the phrase: "...could react with oxygen in its singlet excited state" throughout the paper, including the abstract. I believe that it is benzo[a]pyrene in its singlet excited state, rather than oxygen (which is in its triplet ground electronic state). This has to be modified, otherwise it completely changes the meaning.
There are a few places where the English can be improved, such as
1. Abstract: "can be direct observed" -> directly
2. page 7: "TADF is exclude" -> excluded
3. page 8: "were reported to dependent" -> to be dependent
4. page 8: "after they intercalation" -> their
Reviewer 3 Report
Han and coworkers reported the excited state dynamic study of benzo[a]pyrene upon reacting with oxygen and ctDNA. The time-resolved and transient spectoscopies are well conducted to reveal the fluorescence quenching of benzo[a]pyrene by oxygen and DNA. This manuscript is well written and it can be publised by addressing the following minor issues.
1) The disucssion that exclues the TADF of benzo[a]pyrene is not sufficient. The key point that can rule out the TADF is not the intensity over varied the temperature; it should be the consistent emission wavelength over temperature that we are expecting the phosphorescence peak emerges at low temperature, which phosphorescence is usually lower-energetic than fluorescence.
2) In Figure 2, the new ESA band around 462 nm should be discussed to reveal the origin. It is not corresponding to the absorption or emission. The reverse trend of 462 nm over other ESA peaks should be discussed.
Author Response
please see the attched file.
